# New Complexity-Theoretic Frontiers of Tractability for Neural Network Training

**Cornelius Brand**
Algorithms & Complexity Group
Vienna University of Technology
Favoritenstraße 9-11, 1040 Vienna, Austria
cbrand@ac.tuwien.ac.at

**Robert Ganian**
Algorithms & Complexity Group
Vienna University of Technology
Favoritenstraße 9-11, 1040 Vienna, Austria
rganian@gmail.com

**Mathis Rocton**
Algorithms & Complexity Group
Vienna University of Technology
Favoritenstraße 9-11, 1040 Vienna, Austria
mrocton@ac.tuwien.ac.at

## Abstract

In spite of the fundamental role of neural networks in contemporary machine learning research, our understanding of the computational complexity of optimally training neural networks remains incomplete even when dealing with the simplest kinds of activation functions. Indeed, while there has been a number of very recent results that establish ever-tighter lower bounds for the problem under linear and ReLU activation functions, less progress has been made towards the identification of novel polynomial-time tractable network architectures. In this article we obtain novel algorithmic upper bounds for training linear- and ReLU-activated neural networks to optimality which push the boundaries of tractability for these problems beyond the previous state of the art. In particular, for ReLU networks we establish the polynomial-time tractability of all architectures where hidden neurons have an out-degree of 1, improving upon the previous algorithm of Arora, Basu, Mianjy and Mukherjee. On the other hand, for networks with linear activation functions we identify the first non-trivial polynomial-time solvable class of networks by obtaining an algorithm that can optimally train network architectures satisfying a novel data throughput condition.

## 1 Introduction

Neural networks are a prominent tool in contemporary machine learning, one which has found ubiquitous applications throughout modern computer science (Goodfellow et al., 2016). A neural network (cf. Figure 1) can be thought of as a directed acyclic network consisting of $n$ sources (typically called *input nodes*), and $w$ remaining nodes, which we partition into *output neurons* (the sinks) and *hidden neurons* (all other nodes in the network, typically organized into *layers*).

Given the prominence of neural networks, it is surprising that—in spite of recent efforts—relatively little is known about complexity-theoretic upper bounds for the funda-

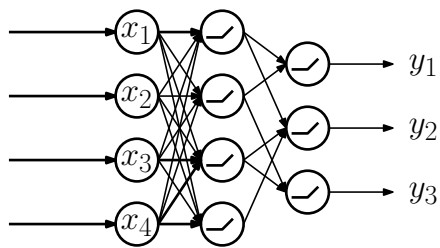

Figure 1: A neural network with two layers of hidden, ReLU-activated neurons, computing a function $f : \mathbb{R}^4 \to \mathbb{R}^3$.

37th Conference on Neural Information Processing Systems (NeurIPS 2023).

mental problem of Neural Network Training (NNT): given a network and a training data set containing $m$ samples, compute weights and biases that best fit the training data.[1] This contrasts recent advances leading to novel algorithmic upper bounds as well as extensive complexity-theoretic landscapes for many other problems arising in machine learning and neural network research, including, e.g., principal component analysis (Simonov et al., 2019; Dahiya et al., 2021), clustering (Ganian et al., 2022), Bayesian network learning (Ordyniak & Szeider, 2013; Ganian & Korchemna, 2021; Grüttemeier & Komusiewicz, 2022) and matrix completion (Ganian et al., 2018).

Naturally, the complexity of solving NNT to optimality strongly depends on the activation function used in the network as well as on what kind of restrictions are placed on the structure of the network, but—as we will see—even in the best studied and simplest cases we lack an understanding of the frontiers of polynomial-time tractability. The Rectified Linear Unit (ReLU) activation function is a natural first choice to consider for a complexity-theoretic investigation; over the past years, it has become the most popular and widely studied activation function used in neural networks (Gao et al., 2020). Goel, Klivans, Manurangsi and Reichman (2021) recently established that the Neural Network Training problem with ReLU activation functions (RELU-NNT) is NP-hard, even when restricted to networks with no hidden neurons, meaning that RELU-NNT is computationally intractable when no further restrictions are placed on the network. In fact, an even more recent reduction shows that RELU-NNT is also complete for the complexity class $\exists \mathbb{R}$ when restricted to complete networks (Bertschinger et al., 2022) with precisely two input nodes and two output neurons, as opposed to networks with arbitrary structure. Moreover, the problem was shown to be hard for the complexity class $\mathsf{W}[1]$ (Froese et al., 2022) when parameterized by $n$ and also intractable for instances of fixed dimension Froese & Hertrich (2023).

While these results seem discouraging at first, an immediate question that arises is whether we can at least efficiently train "small" networks? In particular, what is the complexity of RELU-NNT (as well as other variants of NNT) when restricted to networks where $n$, $w$, or both are (fixed but arbitrary) constants? Here we find a stark contrast between the extensive body of research on the complexity of other natural meta-problems[2] and how little is known for (RELU-)NNT. Indeed, while recent reductions immediately rule out polynomial-time tractability for networks with constant $w$ (Abrahamsen et al., 2021; Froese et al., 2022) or constant $n$ (Bertschinger et al., 2022), the complexity of RELU-NNT in the base case where both $n$ and $w$ are bounded by a constant (i.e., when training constant-size networks) is a prominent open question in the field. The best partial answer we had so far for this high-profile question lies in the seminal work of Arora, Basu, Mianjy and Mukherjee (2018) (see also the follow-up work of Boob, Dey and Lan (2022)), who developed a polynomial-time algorithm that can solve RELU-NNT for constant-size networks with a single layer of hidden neurons and one output neuron.

Naturally, the study of NNT is also relevant for other popular activation functions such as the Sigmoid or Tanh—however, these seem even less amenable to contemporary algorithmic techniques than ReLUs. On the other hand, there exists a class of fundamental activation functions for which the training problem turns out to be easier than for ReLUs: linear functions. Linear-activated networks have been considered in numerous settings as well as theoretical examples (Abrahamsen et al., 2021; Cowley & Pillow, 2020; Panigrahi et al., 2020) and can in many ways be viewed as the "baseline" choice for considering the complexity of neural network training. Indeed, unlike for RELU-NNT, it is not difficult to show that LIN-NNT (i.e., NNT with linear activation functions) is polynomial-time tractable for any network of bounded (i.e., constant) size, but is known to be $\exists \mathbb{R}$-complete for general networks (Abrahamsen et al., 2021) (see also Section 3). However, the complexity of LIN-NNT for general networks with bounded $w$—i.e., with constantly many hidden and output neurons—remains open.

**Contributions.** While several papers have recently provided lower bounds for NNT (Abrahamsen et al., 2021; Froese et al., 2022; Bertschinger et al., 2022), much less progress has been made towards pushing the frontiers of tractability of the problem. In this paper, we remedy this by providing new algorithmic upper bounds that supersede the previous state of the art for RELU-NNT and LIN-NNT.

---

[1] Formal definitions are provided in the Preliminaries.

[2] Consider, e.g., Lenstra's celebrated theorem for solving Integer Linear Programs with constantly-many variables (Lenstra & Jr., 1983) and the multitude of improvements of that initial result (Kannan, 1987; Frank & Tardos, 1987; Ganian & Ordyniak, 2019; Brand et al., 2021), or Schaefer's famous dichotomy theorem for Boolean CSPs (Schaefer, 1978) and its more recent generalization to bounded-domain CSPs (Bulatov, 2017).

As our first major contribution, we establish the polynomial-time tractability of RELU-NNT for all constant-size networks satisfying the property that all hidden neurons have out-degree at most 1. We remark that while this condition is trivially satisfied by every neural network that can be handled by the aforementioned algorithms (Arora et al., 2018; Boob et al., 2022), our result may also be applied to much more general networks and in particular can support more than one output neuron and multiple hidden layers. The algorithm underlying our result can also deal with networks combining ReLU and linear activation functions. Moreover, its running time matches the recent algorithmic lower bound of Froese et al. (2022) for single-neuron architectures, meaning that it is essentially optimal in that setting.

One remarkable consequence of our exact algorithm for RELU-NNT is that it allows us to formulate a procedure which can effectively deal with *every* constant-size network architecture (even those not satisfying the restriction on the out-degree of hidden neurons). In particular, we show that every constant-size ReLU-activated network architecture can be transformed into a new network architecture which is (1) at least as expressive as the original architecture, and (2) can be trained in polynomial time. Crucially, the depth of the new architecture remains the same and its size depends solely on the size of the original architecture.

Next, we turn to the simpler setting of training neural networks with linear activation functions, i.e., LIN-NNT. Recall that here, we are dealing with networks containing a bounded number of hidden/output neurons, but a potentially large number of input nodes. Once we depart from the case of architectures with complete connections between consecutive layers (which can be solved by a direct application of linear regression), training neural networks to optimality on a given data set is known to be hopelessly hard (Abrahamsen et al., 2021) unless the admissible architectures/data are restricted to highly specialized scenarios, such as those with a single output neuron (see Section 3). As our second contribution, we identify a general data throughput condition that allows us to circumvent the aforementioned intractability and guarantee the polynomial-time tractability of LIN-NNT. In particular, we say that a network admits an *untangling* if a subset of its nodes can be partitioned into connected blocks—one for each output and input node—such that:

1a. each input node is the sole source of an input-block;
1b. each output node is the sole sink of an output-block; and
2. the adjacencies between blocks reflects reachability between inputs and outputs.

Intuitively, an untangling can be seen as a backbone in a network where each block either aggregates information from a source or into a sink. While it may not be obvious from the definition at first glance, not only do single-output (as well as single-input) neural networks admit a trivial untangling, but so do many other network architectures one may consider. To exploit our notion of untanglings, we obtain an algorithm that solves LIN-NNT in polynomial time whenever the network comes with an untangling. We combine this result with a procedure that computes an untangling (or determines that none exists) in linear time on all architectures with a constant number of hidden neurons, and provide a different linear-time procedure to find untanglings on architectures of constant *treewidth* (Robertson & Seymour, 1983; Bodlaender, 2016).

## 2 Preliminaries

**Graphs.** We assume basic knowledge of graph terminology (Diestel, 2012). The graphs considered in this paper are assumed to be *directed*, that is, a *graph* is a pair $G = (V, E)$, where $V$ is a set of vertices and $E$ a set of directed edges. For a vertex $v$, the vertices $u$ such that there is an edge $(u, v)$ in $E$ are called the *in-neighbors* of $v$; correspondingly, the set of in-neighbors is the *in-neighborhood* of $v$. The notions of of out-neighbors and out-neighborhood are defined symmetrically. The *in-degree* and *out-degree* of a vertex is the size of its in- and out-neighborhood, respectively. For a vertex subset $X \subseteq V$, we denote by $G[X]$ the graph induced on $X$.

**Neural Networks.** The protagonists of this article are neural networks, which are acyclic graphs endowed with additional information. In particular, the *architecture* of a neural network $\mathcal{N}$ is a directed acyclic graph $G = (V, E)$ where the nodes are called *neurons*. We refer to nodes with no incoming edges as *input nodes*, while those with no outgoing edges are called *output neurons*. All other nodes are called *hidden neurons*. We refer to edges as *deep* when they are not incident to an input node. Typically, one assumes that the hidden neurons are partitioned into *layers* defined by their

shortest distance to an input node, where a hidden neuron in layer $i$ only has in-neighbors in layer $i - 1$ and out-neighbors in layer $i + 1$.

In addition to its neural structure, $\mathcal{N}$ is defined by assigning a *bias* $b_v \in \mathbb{R}$ for each non-input neuron $v$ and a *weight* $a_e \in \mathbb{R}$ for each edge $e$. Furthermore, we fix an *activation function* $\sigma : \mathbb{R} \to \mathbb{R}$ for the entire neural network. To formally map the data dimensions onto the input and output nodes in an architecture, we will assume that these nodes come with an implicit predetermined ordering.

A neural network defined as above *computes* a function $f_\mathcal{N} : \mathbb{R}^n \to \mathbb{R}^d$, where $n$ is the number of input nodes, and $d$ the number of output neurons of $\mathcal{N}$. To define $f_\mathcal{N}$, we say for each neuron $v$ that it computes a function $f_v : \mathbb{R}^n \to \mathbb{R}$, which is inductively defined as follows: The function computed at the $i$-th input node is the coordinate projection $\pi_i : \mathbb{R}^n \to \mathbb{R}, (x_1, \ldots, x_n) \mapsto x_i$. For a non-input neuron, let $v_1, \ldots, v_t$ be the in-neighborhood of $v$, again in some arbitrary, but fixed order, and let $\mathbf{a}_v$ be the vector of weights $(a_{v_i v})_{i=1,\ldots,t}$ in the same order. Furthermore, let $\mathbf{y}_v = (y_1, \ldots, y_t)$ be the values of the functions computed at the neuron $v_i$, respectively—that is, $y_i = f_{v_i}(x_1, \ldots, x_n)$. Then, $v$ *computes* $\sigma(\mathbf{a}_v^T \cdot \mathbf{y}_v + b_v)$. The function computed by the neural network $\mathcal{N}$ itself is then given by $(f_{o_1}(x_1, \ldots, x_n), \ldots, f_{o_d}(x_1, \ldots, x_n))$, where $o_1, \ldots, o_d$ are the output neurons of $\mathcal{N}$.

Clearly, the function computed by $\mathcal{N}$ depends, among other things, on the employed activation function. Here we will consider two specific and well-studied choices for $\sigma$, namely *ReLU* and *linear* activations. It is well known that, without loss of generality, these can be assumed to be $\sigma(x) = x$ (i.e., the identity) for linear activations and $\sigma(x) = \max\{0, x\}$ for ReLU activations (Goodfellow et al., 2016). We will also employ the notation $x^+ = \max\{0, x\}$.

**Training Neural Networks.** The task of *training* a neural network is to, given an architecture and activation, compute a set of weights as well as biases, such that the function computed by the resulting neural network approximates in the best possible manner a given set of training data, that is, a given set of samples $(x, y)$ where $x$ and $y$ are vectors of the appropriate dimension for the network architecture.

More formally, the input to the $\sigma$-NNT problem is an architecture $G = (V, E)$ and data points $D \subseteq \mathbb{R}^n \times \mathbb{R}^d$, where $n$ is the number of input and $d$ the number of output neurons of the architecture. For every setting $a$ of all edge weights as well as $b$ of all biases, this fully determines a neural network $\mathcal{N}_{a,b}$ that in turn computes a function $f_{a,b} = f_{\mathcal{N}_{a,b}}$. The task is then to output a weight $a_e$ for every edge $e$ in $G$ as well as a bias $b_v$ for every neuron such that the function computed by the resulting neural network $\mathcal{N}_{a,b}$ minimizes, among all such choices of $a$ and $b$, the $\ell_2^2$-loss function $L_D(a,b) = \sum_{(x^*, y^*) \in D} ||y^* - f_{a,b}(x^*)||_2^2$, where $|| \cdot ||_2$ is the standard Euclidean norm.

Note that the problem formalization in particular entails that we do not regard the type of activation function as part of the input, whereas the network architecture *is* part of the input by default. This either matches or generalizes previous formalizations of the problem (Arora et al., 2018; Abrahamsen et al., 2021; Goel et al., 2021; Froese et al., 2022; Bertschinger et al., 2022). We will denote the problems of training neural networks for linear and ReLU activations as LIN-NNT and RELU-NNT, respectively, and the analogous problem where the network contains both linear and ReLU activation functions as MIXED-NNT. Furthermore, while the definition of $\sigma$-NNT implies that a *solution* is one which minimizes the $\ell_2^2$-loss function, we will also sometimes call these "optimal solutions" for emphasis. Last but not least, throughout the article we assume that $|D| \geq 2$ since the case where $|D| \leq 1$ is trivial.

## 3 Synopsis of Known Results

Neural network training is a vast area of research that has received an immense amount of attention both on the theoretical as well as the empirical side. In order to put our results into context, we survey the complexity-theoretic understanding of linear and ReLU neural network training.

**Linear Activation.** In the simplest case of LIN-NNT where we assume the size of the whole architecture to be upper-bounded by a constant, it is known that the problem can be expressed as a constant-size system of polynomial inequalities which can be solved in polynomial time, e.g., via the seminal work of Tarski (1951).

On the other hand, LIN-NNT is known to be intractable even if the number of output neurons is upper-bounded by a fixed constant; in fact, the problem was shown to be ∃R-hard even when the number of output neurons is fixed to 3 (Abrahamsen et al., 2021).

One intermediate case between the general and fixed-size fixed-dimension networks is when the network architecture is unrestricted in size but has a special structure, namely, where all edges between layers of nodes exist. In this case, LIN-NNT can be solved in polynomial time by a direct application of multidimensional linear regression (Velu & Reinsel, 2013).

The preceding discussion leaves open a variety of possible directions, such as the complexity of training networks that are not necessarily fully connected, but constrained to have two or even a single output neuron. On the other hand, one may ask what happens if we consider only the number of hidden neurons to be bounded by a constant, without restricting the number of input nodes and output neurons. Our results deal with an intermediate case that captures, among other things, the easy case of single-output networks, as well as some more general networks with a constant number of hidden neurons. In particular, we will formulate a technical condition—whether or not the architecture can be *untangled*—that will allow us to train such networks with linear activations in polynomial time.

**Rectified Linear Activation.**    Recent results from computational complexity theory provide strong lower bounds on the inherent difficulty of the problem of neural network training with ReLU activation functions, already on highly restrictive architectures. Indeed,Goel et al. (2021) and independently Dey et al. (2020) have proven that, under standard complexity assumptions, even training a single ReLU-activated neuron with $n$ input nodes cannot be accomplished in time polynomial in $n$ and the size of the training set $D$; see also the related lower bounds of Bertschinger et al. (2022).

A seminal algorithmic result is that of Arora et al. (2018) for training shallow networks consisting of one layer of hidden ReLU neurons and a single output neuron equipped with a linear activation function. Their algorithm runs in time $|D|^{\mathcal{O}(w \cdot n)}$, where $D$ is the training set, $n$ the number of input nodes and $w$ the number of other nodes in the architecture. Boob et al. (2022) showed that the same approach can be used to handle cases where the output neuron is equipped with a ReLU activation function as well. Froese et al. (2022) have shown that this running time dependency on $n$ in the exponent of $D$ is asymptotically optimal even for a single hidden neuron, ruling out algorithms running in time, e.g., $2^n \cdot D^{o(n)}$ for this basic case.

It is a major open problem to extend the techniques of Arora et al. (2018) to general deep networks—in particular, the complexity of RELU-NNT when restricted to networks whose size is upper-bounded by an arbitrary but fixed constant is wide open. Our work shows that it is nevertheless possible to identify a much more general set of architectures than those covered by the aforementioned results (Arora et al., 2018; Boob et al., 2022) for which RELU-NNT is polynomial-time solvable.

## 4   Training ReLU Networks

In this section, we establish our tractability results for ReLU-activated neural network training when each hidden neuron has out-degree at most $1$. As our first step, we prove that in such instances the weights in the hidden layers can be discretized to only two values.

**Lemma 1.** *For any ReLU network $\mathcal{N}$ and for any hidden neuron $u$ with precisely one out-neighbor $w$, there is a ReLU network $\mathcal{N}^*$ with the same architecture such that $f_{\mathcal{N}} = f_{\mathcal{N}^*}$ and $a_{uw}^* \in \{-1, 1\}$. Moreover, the only parameters in which $\mathcal{N}$ and $\mathcal{N}^*$ differ are the weights of the edges incident to $u$ and the bias of $u$.*

*Proof Sketch.* Let $(v_i)_i$ be the (non-empty) family of predecessors of $u$, and $w$ its only successor. To obtain $\mathcal{N}^*$, we construct a new set of weights and biases at $u$, where $\mathrm{sgn}(x)$ is the sign function:

$$\forall i, a_{v_i u}^* = a_{v_i u} \cdot |a_{uw}|$$
$$b_u^* = b_u \cdot |a_{uw}|$$
$$a_{uw}^* = \mathrm{sgn}(a_{uw}) \quad (1 \text{ if } a_{uw} = 0)$$

All other weights in $\mathcal{N}^*$ remain the same as in $\mathcal{N}$. To complete the proof, it suffices to show that $f_{\mathcal{N}} = f_{\mathcal{N}^*}$ holds by using the fact that the ReLU function commutes with multiplication with non-negative factors. □

By iteratively applying Lemma 1, we obtain:

**Lemma 2.** *Let $\mathcal{N}$ be a ReLU structure such that every hidden neuron has out-degree 1. Then there exists an optimal solution to* ReLU-NNT *such that all deep edges have weights in* $\{1, -1\}$.

Lemma 2 will later allow us to reduce our search space for ReLU-NNT by only needing to consider two options for the weights of each deep edge. We note that Maass (1997) employed a related idea.

**Partitioning by Hyperplanes.** The other key argument used to reduce the search space further concerns the way ReLU neurons partition the training set into a "dead area" (i.e., the elements where the linear unit is rectified) and an "active area" (where the unit behaves as a linear function). More precisely, given a data set $D$ and a ReLU-activated neuron $u$, we say that $u$ *partitions* $D$ into an *active area* $D|_u$, which contains all data points for which $f_u$ outputs a value greater than zero, and a *dead area* $\overline{D|_u} = D \setminus D|_u$, which contains all data points for which $f_u$ outputs zero.

As a base case, let us consider a neuron $u$ in the first hidden layer of a neural network and let $\mathbf{a}_u^T$ be the vector of weights for the edges from the input nodes to $u$, where a non-edge would be represented as a weight of 0. As a trivial upper bound, there are in general at most $2^{|D|}$ ways $u$ can partition $D$ into an active and dead area, which provides a trivial upper bound for an algorithm enumerating all such partitions. However, we have that $f_u(D_i) = (\mathbf{a}_u^T \cdot D_i + b_u)^+$ for each data point $D_i \in D$ and moreover $\mathbf{a}_u^T \cdot \mathbf{x} + b_u$ defines a hyperplane in an $n$-dimensional space. This allows us to obtain an algorithm with a better running time bound for enumerating the partitions of the point set that is inspired by a result of Megiddo (1988).

**Lemma 3.** *Let $D \subseteq \mathbb{R}^n$ be a finite set of points. Then, the set of partitions $\Pi = \{(A, B) \mid A \cup B = D$ and $A, B$ are separated by a hyperplane$\}$ can be enumerated in time* $|D|^{\mathcal{O}(n)}$.

*Remark.* A similar result was claimed and used by Arora et al. (2018, Page 14) in their algorithm for dealing with ReLU-NNT restricted to a special case of the setting treated in this section. However, the argument presented there seems to be incomplete. In particular, it makes the claim that "the total number of possible hyperplane partitions of a set of size $D$ in $\mathbb{R}^n$ is at most $2\binom{D}{n}$", which is imprecise (consider, e.g., $|D| = n$). Moreover, the book listed as reference for bounding the total number of possible hyperplane partitions does not provide an algorithm for enumerating these efficiently, and enumeration is required for both our result and the result claimed in Arora et al. (2018). A bound on the number of such partitions can be attributed to Harding (1967), but that does not yield efficient enumeration either. If the points are known to lie in general position, efficient enumeration could be carried out via a translation to hyperplane arrangements followed by an application of the results of Edelsbrunner et al. (1986); however, in our setting we cannot guarantee that these lie in general position, necessitating our stand-alone proof of Lemma 3 above (which we believe to be also of general interest).

The considerations preceding Lemma 3 do not immediately translate to neurons that are deeper than in the first layer, since the inputs to these activation functions are obtained by non-linear transformations of $D$. Nevertheless, we prove that it is also possible to derive an upper bound for neurons beyond the first hidden layer if we are provided information about the partitions of neurons in previous layers.

**Lemma 4.** *Let $D$ be a set of data points, $u$ be a ReLU neuron, $F$ be the set of all neurons on paths from input nodes to $u$, and $x = |F|$. Given $D|_v$ for each $v \in F$, we can upper-bound the number of distinct active areas $D|_u \subseteq D$ over all networks consistent with the given selection of $\{D|_v \mid v \in F\}$ by $|D|^{\mathcal{O}(n \cdot 2^x)}$, and it is possible to enumerate a superset of these in time $|D|^{\mathcal{O}(n \cdot 2^x)}$.*

We now show how to solve ReLU-NNT in polynomial time for each fixed value of $n$ and $w$:

**Theorem 5.** *There is an algorithm that, given an instance $(G, D)$ of* ReLU-NNT *such that every hidden neuron of the structure has out-degree exactly 1, and the data points are encoded using $L$ bits overall, computes the global optimal solution in time* $|D|^{\mathcal{O}(n \cdot w \cdot 2^w)} \cdot \text{poly}(L)$.

*Proof Sketch.* The algorithm begins by exhaustively branching to determine the weight of each deep edge, which requires us to consider at most $2^w$ options in view of Lemma 2. Once the weights of the deep edges are fixed, we branch to determine how each hidden and output neuron partitions the training set. For this to work, we need to process layers consecutively—we only branch on the possible partitions for one neuron after we have chosen the partitions for all of its hidden in-neighbors.

For each such neuron, we enumerate all feasible partitions of $D$ using Lemma 4, and branch over the possibilities. For a given partition, we create a set of inequalities that are satisfied only if each element of $D$ indeed is affected by $u$ according to the partition. Once a partition has been chosen for all neurons, the function computed by the resulting neural network $\mathcal{N}$ for each neuron is fixed save for the weights of edges adjacent to input nodes, and the biases. We will optimize over all these free variables by constructing an instance of the QUADRATIC OPTIMIZATION problem, whereas we optimize $L_D(a, b)$ subject to the set of constraints obtained when partitioning the neurons. The obtained instances of QUADRATIC OPTIMIZATION can be solved in time polynomial in the number of variables and constraints, as well as the bitlengths of the datapoints, e.g., by using the ellipsoid method (Pang, 1983; Kozlov et al., 1979). $\qquad\square$

Note that the proof of Theorem 5 can be directly extended to also include architectures where some of the hidden or output neurons use a linear activation function instead of ReLU. Indeed, we need not guess a partition for the neuron and hence not construct a constraint in the quadratic program, and furthermore we may treat a linear neuron the same way as an activated ReLU in the following layer. This yields the following theorem, which generalizes the tractability results of both Arora et al. (2018) and Boob et al. (2022).

**Corollary 6.** *There is an algorithm that, given an instance $(G, D)$ of* MIXED-NNT *such that every hidden neuron of the structure has out-degree $1$, and the data points are encoded using $L$ bits overall, computes the global optimal solution in time $|D|^{\mathcal{O}(n \cdot w \cdot 2^w)} \cdot \mathrm{poly}(L)$.*

## 5 Blowing Up a Neural Network

The tractability results of Arora et al. (2018) on constant-width shallow ReLU-networks raise the natural question of whether it is possible to (efficiently) perform depth reduction on deeper networks, so as to reduce a given (deep) instance for RELU-NNT to a tractable (e.g., shallow) special case. It is widely conjectured that this is not possible in general without sacrificing some quality of the solution (see, e.g., Hertrich (2022), Conjecture 3.1. and Haase et al. (2023); Hertrich et al. (2021)). Among others, it is widely believed that there are functions that are efficiently computable by depth-3 but not by depth-2 ReLU-activated neural networks (Eldan & Shamir, 2016), and similar results were also shown for deeper networks (Cohen et al., 2016; Telgarsky, 2016).

While these results essentially rule out performing depth reduction without a negative impact on the accuracy, our tractability results in Section 4 allow us to approach the question from a different angle. In particular, we show that it is possible to transform an arbitrary constant-size ReLU-activated deep neural network architecture into one of the same depth which is (1) at least as powerful as the original architecture, (2) guaranteed to admit polynomial-time training, and (3) upper-bounded in size by a function of the size of the original architecture[3].

To explain the construction, we start again with the base case of a single neuron. More precisely, let $v$ be a single ReLU-activated (hidden) neuron with $n$ inputs and $d$ outputs, and say the weights on the input edges are $a_1, \ldots, a_n$, the weights on the output edges are $a'_1, \ldots, a'_d$, and the bias at the hidden neuron is $b$. Now consider a new network with a single layer of ReLU-neurons which consists of $d$ ReLU-activated neurons $v_1, \ldots, v_d$, where every $v_i$ is connected to all inputs with the same weights $a_1, \ldots, a_n$ as before. Furthermore, each $v_i$ has a single output edge with weight $a'_i$. We call this network the *blow-up* of $N$. Let us first establish that this blow-up has the desired property:

**Lemma 7.** *Let $N$ be a neural network with a single ReLU-activated hidden neuron as before, and let $N'$ be its blow-up. Let $f_N, f_{N'} : \mathbb{R}^n \to \mathbb{R}^d$ be the functions computed by $N$ and $N'$, respectively. Then, $f_N = f_{N'}$ holds.*

This procedure can be generalized to blow up an arbitrary ReLU-activated neural network, regardless of the number of its layers.

With a slight abuse of notation, we will also refer to the *blow-up* of a given network architecture $G$ (that is, one without weights and biases) as the architecture of the blow-up of an arbitrary neural network with architecture $G$, and the process for obtaining the blow-up of an architecture is the very

---

[3]We remark that such a transformation can be trivially provided for linearly-activated network architectures, since LIN-NNT is polynomial-time solvable on complete networks. We also explicitly note that changing the architecture is not always desirable or even possible.

same as for a weighted network (ignoring any weights). We now turn to establishing the properties of blow-ups with respect to RELU-NNT.

**Proposition 8.** *Let $G$ be an arbitrary architecture and let $G'$ be its blow-up. Furthermore, let $L_D$ and $L'_D$ be the loss functions associated with $G$ and $G'$, respectively. Then, for any weights $a, b$ on $G$ and the minimizer $a^*, b^*$ of $L'_D$ on $G'$, we have $L'_D(a^*, b^*) \leq L_D(a, b)$.*

The size of the blow-up of a network with a constant number of hidden neurons behaves as follows.

**Proposition 9.** *Let $\mathcal{N}$ be a neural network with $\ell$ layers of $\lambda \geq 2$ hidden neurons each, and $o$ output neurons. Then the blow-up of $\mathcal{N}$ has at most $b \leq o \cdot \lambda^{\ell+1}$ hidden neurons, and at most $o \cdot \lambda^\ell$ neurons in its first hidden layer. If $\lambda = 1$, we have $b \leq o \cdot \ell$.*

Recalling Theorem 5 from Section 4, we now directly obtain:

**Theorem 10.** *There is an algorithm that*

- *takes as input an instance $(G, D)$ of RELU-NNT where $G$ has depth $\ell$, $n$ input nodes, $o$ output neurons and at most $\lambda$ hidden neurons per layer,*
- *computes the blow-up $G'$ of $G$ and an optimal solution to RELU-NNT on $(G', D)$, and*
- *runs in time $|D|^{\mathcal{O}(n \cdot \tau \cdot 2^\tau)}$, where $\tau = (\ell + \lambda^{\ell+1}) \cdot o$.*

Observe that the running time in Theorem 10 is polynomial for every architecture of constant size and essentially tight with respect to the lower bound of Froese et al. (2022) for architectures with a single hidden neuron. In combination with Proposition 8, this implies a polynomial-time algorithm which transforms an arbitrary constant-size architecture into a new one which has already been solved to optimality, with no sacrifice in either accuracy or depth.

We remark that the procedure described in this section can be seen as a rough analogue to *improper learning* in learning theory in that it involves a change of the hypothesis class. In particular, while we prove that our blow-up procedure does not increase the training error, it might increase the generalization error of empirical risk minimization in learning settings. We also point out that a similar procedure was described in an earlier work of Maass, albeit in it was applied in a different context Maass (1997).

## 6  Training Linear Networks

**Untanglings of Linear Networks.**   As discussed in Section 3, the complexity of LIN-NNT is open when restricted to architectures with arbitrarily many input nodes while considering the number of other nodes in the network to be fixed by a constant, and very few classes of architectures are known to admit polynomial-time training for linear activations. Here, we identify a general substructure of the architecture which, when present, guarantees the polynomial-time solvability of LIN-NNT.

Specifically, an *untangling* of an architecture $G = (V, E)$ with input nodes $x_1, ..., x_\alpha$ and output nodes $y_1, ..., y_\beta$ is a partitioning $\Pi = (B, C_1, ...C_\alpha, D_1, ..., D_\beta)$ such that:

- Each $G[C_i]$, $1 \leq i \leq \alpha$ contains $x_i$ as its only source;
- Each $G[D_j]$, $1 \leq j \leq \beta$ contains $y_j$ as its only sink; and
- For each $x_i$ and $y_j$, if $y_j$ is reachable from $x_i$ in $G$, there exist $s \in C_i$ and $t \in D_j$ such that $st \in E$.

As the first result in this section, we will show that if one is given a network architecture along with an untangling, it is possible to solve LIN-NNT in polynomial time. Towards this, we observe that since all transformations in such networks are affine, it is possible to assume without loss of generality that non-zero biases are only present on output neurons. Indeed, one can slide the biases along the edges in the network towards the output neurons, and if we multiply them by the weights of edges met along the way, the computed function does not change. As an example, given an edge with weight $a_1$ from a neuron associated with bias $b_2 \neq 0$ to a neuron associated with bias $b_1$, we can alter the biases by setting $b_2 := 0$ and $b_1 := b_1 + a_1 \cdot b_2$ since $a_1 \cdot (x + b_2) + b_1 = a_1 \cdot x + (b_1 + a_1 \cdot b_2)$. Applying this iteratively along all the edges of a network yields:

**Observation 11.** *For each neural network $\mathcal{N}_{a,b}$ with linear activation functions, there exists a neural network $\mathcal{N}_{a,b'}$ such that $b'$ is non-zero only on the output neurons and $f_{\mathcal{N}_{a,b}} = f_{\mathcal{N}_{a,b'}}$.*

Observation 11 allows us to proceed towards establishing the claimed tractability result.

**Theorem 12.** LIN-NNT *can be solved in time* $\mathrm{poly}(|D| + |V|)$ *if an untangling of the architecture* $G = (V, E)$ *is provided as part of the input.*

**Computing Untanglings.** Given Theorem 12, the natural next question concerns the complexity of actually computing an untangling in a neural network. The aim of this section is to provide a comprehensive answer to that question. As our first result, we prove that deciding whether a given architecture has an untangling is, in general, intractable.

**Theorem 13.** *Deciding whether a given architecture $G$ has an untangling is* NP-*hard.*

*Proof Sketch.* We provide a polynomial reduction from the NP-hard DOMINATING SET 3-PARTITIONING problem (Heggernes & Telle, 1998): Given an undirected graph $H = (V, E)$, decide whether its vertex set can be partitioned as $V = A_1 \cup A_2 \cup A_3$ such that $A_1$, $A_2$ and $A_3$ are all dominating sets of $H$. Here, a *dominating set* of a graph is a subset $X$ of its vertices such that every vertex $v$ is either in $X$ itself, or adjacent to some $w \in X$; we say that such $w$ *dominates* $v$.

Let $H = (V, E)$ be an undirected graph. We first describe how to transform $H$ into an architecture $G$ such that $G$ has an untangling if and only if $H$ can be partitioned into three dominating sets. $G$ is constructed as a three-layered architecture, that is, with vertex set $L_1 \cup L_2 \cup L_3$, where $L_1 = \{d_1, d_2, d_3\}$, $L_2 = V$, and $L_3$ consists of $|V| + 1$ disjoint copies of $V$, that is, $L_3 = \bigcup_{v \in V}\{v^{(1)}, \ldots, v^{(|V|+1)}\}$. Additionally, $G$ has all possible edges from $L_1$ to $L_2$, and $v \in L_2$ has an edge towards $u^{(i)} \in L_3$ in $G$ if and only if $u = v$ or $v$ was adjacent to $u$ in $H$. To complete the proof, it suffices to establish that every untangling in $G$ implies a partitioning of $H$ into three dominating sets, and vice-versa that every such partitioning of $H$ guarantees that there is an untangling in $G$. $\square$

While negative, the intractability of computing an untangling in a general architecture is far from unexpected. For our next result, we take aim at the restriction of LIN-NNT where the architectures have an arbitrary but fixed bound on the number of hidden neurons—a case which is notable not only due to the fact that the target architectures will typically be much smaller than the training set, but also because its complexity remains open (cf. Section 3). We show that under this restriction, the problem of determining whether the architecture has an untangling is solvable in linear time.

**Theorem 14.** *There is an algorithm which either computes an untangling in a given architecture* $G = (V, E)$ *with $k$ hidden neurons in time* $k^{\mathcal{O}(k)} \cdot |V|$, *or correctly determines that none exists.*

*Proof Sketch.* Let $W \subseteq V$ be the set of hidden neurons in $G$. We begin by exhaustively branching over all of the at most $k^{\mathcal{O}(k)}$ partitionings of $W$ into sets $\Pi' = (B, C'_1, \ldots C'_\alpha, D'_1, \ldots, D'_\beta)$ (for all choices of $\alpha$ and $\beta$, both of which are trivially upper-bounded by $k$). For each such choice of $\Pi'$, the algorithm then proceeds by determining whether it can be extended to an untangling of $G$ by adding an input node to each $C'_i$, $1 \leq i \leq \alpha$, an output neuron to each $D'_j$, $1 \leq j \leq \beta$, and keeping all remaining input nodes and output neurons as singletons in the untangling, in a way which satisfies the reachability constraint of an untangling. To this end, we will employ a further set of exhaustive branching rules which will iteratively assign singleton output neurons and input nodes to the sets in $\Pi'$. Intuitively, whenever we identify that an input node is missing a connection to an output node (due to at least one of these not being assigned to a part in $\Pi'$), we exhaustively branch over all ways one of these two neurons can be assigned to a part in $\Pi'$. $\square$

As our final result, in the next Theorem 15 we show that it is also possible to compute untanglings in linear time even on architectures with many hidden neurons, provided that such architectures are "well-structured" in a graph-theoretic sense. In particular, we show that this holds for architectures which are tree-like in the sense of having bounded *treewidth*: a highly established structural graph parameter which measures how tree-like a graph is (Robertson & Seymour, 1983). Specifically, here we consider the treewidth of the underlying undirected graph, i.e., the simple graph obtained by replacing each directed edge in the architecture by an undirected one.

To put Theorem 15 into context, we remark that the proof is based on an application of Courcelle's well-known Theorem (Courcelle, 1990) and hence a definition of treewidth will not be required for an exhibition of the result or its proof; instead, we refer to the multitude of existing surveys and materials on the topic (Courcelle & Engelfriet, 2012; Bodlaender, 2016). We remark that the result is incomparable to Theorem 14: while it can easily be shown that every architecture with $k$ hidden

neurons also has treewidth at most $k$ and hence Theorem 15 can be applied to a strict superclass of architectures, the running time guaranteed here has a nonelementary dependency on the treewidth of the graph (as opposed to the $k^{\mathcal{O}(k)}$ dependency in Theorem 14).

**Theorem 15.** *Let $t$ be an arbitrary but fixed integer. Given an architecture $G = (V, E)$ of treewidth at most $t$, it is possible to compute an untangling (or correctly determine that none exists) in time $f(t) \cdot |V|$ for some computable function $f$.*

## 7 Concluding Remarks

We have examined the theoretical boundaries of computational tractability for training neural networks both for the fundamental cases of ReLU and linear activations, complementing a flurry of recent results establishing increasingly stronger lower bounds for network training. Our results generalize recent algorithms for optimal ReLU-activated network training (Arora et al., 2018; Boob et al., 2022) and are among the first to identify a non-trivial class of linear-activated architectures admitting polynomial-time training.

A natural direction for future work is to examine the extent to which the identified islands of tractability can be generalized before stepping into intractable territory. A long-term goal is to obtain a clear cutoff between intractable and tractable cases of network training. We conclude with three specific questions directly arising from our results:

1. Is LIN-NNT polynomial-time tractable when restricted to constant-treewidth architectures?
2. Is it possible to solve RELU-NNT for constant-size architectures containing hidden neurons with out-degree greater than 1?
3. Can the obtained results be generalized to other loss functions?

## Acknowledgments and Disclosure of Funding

The authors confirm that there are no competing interests. We acknowledge support by the Austrian Science Fund (FWF, START Project Y1329). Mathis Rocton further acknowledges support by the European Union's Horizon 2020 research and innovation COFUND programme (LogiCS@TUWien, grant agreement No 101034440).

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
