# OpenReview forum: "New Complexity-Theoretic Frontiers of Tractability for Neural Network Training"
_NeurIPS.cc/2023/Conference — NeurIPS 2023 poster_

### Official Review · Reviewer_boAZ · 2023-06-17

**Soundness:** 4 excellent
**Presentation:** 3 good
**Contribution:** 3 good
**Rating:** 6
**Confidence:** 3

**Summary:**

The paper studies the neural network training problem for ReLU and linear networks. They show new settings where the problem can be solved in polynomial time. They provide two main contributions:
- For ReLU networks, they give a polynomial-time algorithm for training constant-size networks where the out-degree of all hidden neurons is $1$. This extends similar previous results for depth-$2$ networks. They observe that every constant-size network can be transformed into a network with the above structure.
- For linear networks, they identify a condition, called untangling, that allows for efficient learning if the untangling is given. Finding an untangling is NP-hard in general, but if the network has a constant number of hidden neurons or a constant treewidth, then it can be found in linear time.

**Strengths:**

The neural network training problem is a natural question that has been studied in several prior works. This paper makes a further step towards understanding when this problem can be solved efficiently, for both linear and ReLU networks.
Also, the paper is well-written.

**Weaknesses:**

My comments are regarding the result on ReLU networks:
- The result requires out-degree of exactly $1$ for all hidden neurons. It is indeed more general than the results of Arora et al. 2018 and Boob et al. 2022, but the family of networks covered by the new result and not by the previous ones is somewhat restricted. The authors show that every constant-size network can be transformed into a network with the above structure, but it is not clear to me what the implications of the ability to solve the problem for the transformed network are. It would be helpful if the authors elaborated further on the motivation for this transformation.
- I am trying to understand what are the main technical differences between the current result and Arora et al. 2018, and what are the main non-trivial steps. I would be happy if the authors elaborate more on this issue. Below I provide some comments in this respect:
    - Regarding Lemmas 1 and 2: The idea of “pushing the weights backward” in ReLU networks with out-degree $1$ of each neuron (i.e., making all weights in {-1,0,1} in all layers except for the first layer) is not new. It was done, e.g., in [1].
    - Regarding Lemma 3: The authors compare it to Arora et al. in the Remark, and claim that both the bound on the number of partitions and the efficient enumeration method are essentially new. I actually believe that the $|D|^{O(n)}$ bound on the number of partitions follows immediately from the Sauer–Shelah lemma (do you agree?). I don’t know whether an efficient enumeration algorithm was previously known. Is it the main technical contribution in this result?
    - Regarding Lemma 4: I believe that the Sauer–Shelah lemma can give a tighter bound here (namely, $|D|^{O(n \cdot \text{poly}(x))}$). Again, I don’t know regarding efficient enumeration.
    - Given Lemmas 1-4, the proof of Theorem 5 becomes similar to the result from Arora et al., right?
    - The result in Section 5, namely, transforming networks to larger ones with out-degree $1$, is a trick that was also used in prior works. As far as I understand, this is exactly Remark 2.2. from [1].
- Typo in line 215: should be $a_{u,w}^*$.

[1] Maass, Wolfgang. "Bounds for the computational power and learning complexity of analog neural nets." Proceedings of the twenty-fifth annual ACM symposium on Theory of computing. 1993.


**Questions:**

See the “weaknesses” section.

**Limitations:**

The authors discussed the limitations.

---

> ### Author Rebuttal · Authors · 2023-08-08
>
> Regarding our results on ReLU networks, the class of networks where our algorithmic upper bound applies can indeed be considered as restrictive - especially when one views it from the perspective of practical applications. However, from a fundamental graph-structural viewpoint, we believe that being able to solve architectures that have an out-degree of 1 is significantly more general than the previously established tractability result for the 3-layer architectures with a single output neuron considered by Arora et al. 2018, and the proof of our result does not straightforwardly follow from the previous state of the art.
>
> As for motivating our result that “every constant-size network can be transformed into a network with the above structure”, we believe this is indeed of significant theoretical interest. Essentially, while the first result identifies a tractable subclass of fixed-size ReLU architectures, this latter result allows us to say something about *every* arbitrary fixed-size ReLU architecture. In particular, while solving NNT on such architectures is intractable in the complexity-theoretic sense, we show that one can transform the architecture into one that is at least as powerful while also being tractable. From a foundational perspective, we believe the existence of such a transformation to be somewhat surprising and not at all obvious. That being said, the result follows as somewhat of a corollary to our first algorithmic result and hence we would be willing to de-emphasize it if the reviewers prefer.
>
> We believe that the main technical challenge that sets apart our first result from the previous result of Arora is that here we had to provide an efficient enumeration algorithm which can be applied inductively even for hidden nodes that lie deeper in the architecture; this can be seen as a “supercharged” version of the result they used in their setting. (As we note in the article, we also put this result on more solid theoretical footing.) While we are aware of several ways one could bound the number of such partitions, our algorithm requires us to efficiently enumerate them. Moreover, we would like to note that the article provides not only novel algorithmic upper bounds for ReLU-NNT but also for Lin-NNT; we believe these to also be an important contribution.
>
> Finally, thank you for the reference to [1]; we will incorporate it into our discussion in the final version of the article. We remark that while their construction is closely related to ours, the results and proofs obtained in [1] do not allow us to skip any of the steps in our proofs.

---

> > ### Comment · Reviewer_boAZ · 2023-08-13
> >
> > Thanks for the response. I increased my score by 1.

---

### Official Review · Reviewer_3t5Y · 2023-06-26

**Soundness:** 4 excellent
**Presentation:** 4 excellent
**Contribution:** 3 good
**Rating:** 7
**Confidence:** 5

**Summary:**

This paper studies the computational complexity of empirical risk minimization (ERM) for neural networks, that is, given an architecture and training data points, find weights and biases of a global minimum of the training error. The problem is well-known to be NP-/ER-/W[1]-hard already in very easy special cases. This paper identifies a bunch of cases in which it can still be solved in polynomial time, namely in each of the following cases:
- for ReLU networks with all hidden neurons having out-degree one, if the size of the NN is regarded as a fixed constant,
- for an improper ReLU learning setting, leading essentially to the case of the first bullet point by blowing up the architecture,
- for linearly activated NNs if they fulfill a certain structural property.

Moreover, the authors analyze the computational complexity of determining whether a linearly activated NN satisfies the structural property mentioned in the third bullet point.

**Strengths:**

- Given the significance and hardness of the problem, finding tractable special cases is a very important contribution to the theory of machine learning. This paper provides a decent starting point (building upon Arora et al. (2018)) and hopefully initiates further research towards this goal.
- The paper is mathematically very well-written. It even cleans up with some vagueness in previous work (see Remark after Lem. 3).
- The paper also is very precise in surveying the previous literature on computational complexity of ERM for NNs.

**Weaknesses:**

- The obtained algorithms are only of theoretical interest because (i) the tractable special cases are rather unrealistic, (ii) the running times have still high dependencies on "fixed" constants, and (iii) this work focusses purely on minimizing the training error and neglects learning / generalization. So the paper should really be seen as fundamental theoretical research rather than pushing the frontiers of what we can do in practice. As such, I think it is of significant value.

Apart from this, I have many minor comments as below. None of them is a reason to downvote for me, but I urge the authors to carefully work on them for the next version:

- My impression is that the abstract is not very informative. Instead of spending too much effort on motivating the results there (see also my next comment), it would be better to be more precise of what this paper actually proves (why not simply state the main results?).
- While I believe the paper makes important contributions (as outlined above), I am quite unhappy with the way the results are pitched, e.g. already in the abstract, and in lines 17-20, and other places where the authors claim that so little is known about the complexity of NN training. Yes, it is true that not many positive results have been achieved for pure empirical risk minimization, but this is for a good reason: the problem simply IS really hard in the worst-case, and this has been studied thoroughly. While I do agree that finding tractable special cases is an important research contribution, I believe that this will always only be possible for very special cases and therefore disagree with staging this as such a "huge" gap in the literature.
- a recent preprint you might not be aware of is: Froese, Hertrich: "Training Neural Networks is NP-Hard in Fixed Dimension" (https://arxiv.org/abs/2303.17045). This includes a multitude of new stronger hardness results for both ReLU (and linear threshold) networks. I think you should cite this in line with other negative complexity results you already cite, like (Arora et al., 2018; Abrahamsen et al., 2021; Goel et al., 2021; Froese et al., 2022; Bertschinger et al., 2022). In particular in line 52, for the constant n case, the Bertschinger et al. (2022) result relies on multi-dimensional outputs. The new preprint suggested above shows that NP-hardness can already be achived with a single-dimensional output (and still only 2-dimensional input).
- line 65: "it is not difficult to show that LIN-NNT is poly-time tractable for any network of bounded size": why is this the case? can you please elaborate? And does it depend on whether the NNs are fully-connected or not?
- lines 68-69: are you referring to arbitrary n but constant w here? If so, please make the arbitrary dependence on n more explicit.
- lines 90 and beyond: are you still talking about fixed size?
- line 278: "save" -> "except"?? (or I completely misunderstand this sentence)
- line 282: the ellipsoid method is NOT polynomial in the number of variables and constraints. In fact, its time complexity depends on the bit-length of the encoding of the input. It is what people call "weakly polynomial" and there is no strongly polynomial time algorithm known, even for the special case of linear programming. This also affects the running time of the overall algorithm. It should be something like $|D|^{\mathcal{O}(nw2^w)}\cdot\mathrm{poly}(L)$, where $L$ is the bit-length of the encoding of all data points. Please fix this everywhere in the paper where you state running times involving a quadratic program (for example, this also applies to Thm. 12).
- line 297: in addition to the cited thesis, you should probably also cite the paper the thesis is based on (Hertrich, C., Basu, A., Di Summa, M., & Skutella, M. Towards lower bounds on the depth of ReLU neural networks. NeurIPS 2021.) and additionally the following paper making significant progress towards proving the conjecture: Haase, C. A., Hertrich, C., & Loho, G. Lower Bounds on the Depth of Integral ReLU Neural Networks via Lattice Polytopes. ICLR 2023.
- line 305: point (2): "polynomial-time training" is very misleading here, this is only true if the architecture size is regarded as a fixed constant!
- general comment about Section 5: what you are doing here is known as improper learning in the learning theory community. You should probably mention the term and compare to the respective literature. In particular, while you prove that your blow-up procedure does not increase the training error, such a move might heavily increase the generalization error of empirical risk minimization in learning settings. You should mention this!
- line 348: Are you a "such that" missing at the end of this line?
- line 379: d3 -> d_3

**Questions:**

No particular questions, but I invite the authors to reply to any of the comments I've made if they disagree.

**Limitations:**

All statements are mathematically rigorous and all assumptions properly stated.

The authors should mention that the blow-up in Section 5 might heavily increase the generalization error in learning settings, see my respective comment above.

The authors should point out more clearly that the purpose of the provided algorithms is purely theoretical, see my first point in "weaknesses".

---

> ### Author Rebuttal · Authors · 2023-08-08
>
> We are grateful for the encouraging (and very helpful) feedback, and completely agree that obtaining an understanding of the fundamental complexity of neural network training is an important part of machine learning. The article improves our understanding of the boundaries of tractability of the problem in the complexity-theoretic sense, and as such the algorithms are - as you correctly mentioned - indeed of theoretical interest.
>
> Please find our responses to the individual minor comments below.
>
> Ad 1) and 2), we will reformulate the abstract and the relevant parts of the introduction in a way which is more informative and better highlights the state of knowledge about the problem (and, among others, we will tone down the claims about the “huge gap” in the literature).
>
> 3) Thank you for the interesting reference - we will incorporate it into our discussion. This also applies to the reference provided in point 9).
>
> 4) This is elaborated on in lines 172-175 (and even more details are provided in the appendix).
>
> 5) Yes indeed; we will make this clearer in the final version of the article.
>
> 6) No, when we switch to Lin-NNT we start by considering the general setting, without any conditions on the input dimension nor the number of hidden (or output) neurons. During our discussion (and specifically on line 109), however, we consider the case of having a bounded number of hidden neurons in order to compute the untangling of an architecture.
>
> 7) We believe both are correct, but will reformulate to “except”.
>
> 8) This is indeed an oversight on our part. We’re assuming a standard Word-RAM model of computation, and the bit complexity does enter in our bounds in the same way that it does for the quadratic optimization methods used; we are grateful to this reviewer and Reviewer SU6D
> for pointing this out. We will of course update the running time bounds in the final version, and remark that this minor change (by a polynomial factor) does not affect the article’s contribution.
>
> 9) See above.
>
> 10) The architecture size (after blow-up) is a function of the initial architecture size, which is fixed, so we indeed are using fixed size architecture here. However, to improve clarity, we will reformulate this sentence to emphasize that the size is fixed.
>
> 11) Thank you, we will mention this.
>
> Ad 12) and 13), both of these typos will be fixed in the final version.

---

> > ### Comment · Reviewer_3t5Y · 2023-08-13
> >
> > I thank the authors for their response, encourage them to incorporate the promised improvements, and continue to vote for acceptance with high confidence.

---

### Official Review · Reviewer_t5Nf · 2023-07-05

**Soundness:** 4 excellent
**Presentation:** 4 excellent
**Contribution:** 2 fair
**Rating:** 6
**Confidence:** 2

**Summary:**

The paper studies algorithms for exact minimization of quadratic loss over ReLU and linear neural networks when the total number of neurons is a fixed constant. The results can be divided into two parts:

* A polynomial time algorithm for a fixed size ReLU/linear network, when it has the following structure: 1) One layer of neurons with arbitrary edges between inputs and those neurons. 2) A tree of neurons (i.e., every neuron has out-degree one) on top of that. More precisely, the algorithm runs in time $|D|^{O(nw 2^w)}$, where $|D|$ is size of the dataset, $n$ number of inputs, and $w$ total number of hidden and output neurons.

* A polynomial time algorithm for a (possibly large) linear network whenever all hidden neurons can be collapsed (like in a graph minor) onto inputs and outputs such that the resulting bipartite graph is equal to the connectivity graph between inputs and outputs. So, it's some sort of condition that says that the graph has a "simple" connectivity structure. This pattern of connectivity needs to be known to the algorithm, the authors prove that it is in general NP-hard to find it.

In both of these settings polynomial-time algorithms are already known for the most natural subcases (for example, one fully connected layer in the ReLU case).

**Strengths:**

* The topic is interesting and relevant. As the authors point out, we know surprisingly little even about simple cases of this basic problem.

* The paper is clearly written, including the context and exhaustive discussion of related work. It was a pleasure to read.

* A lot of attention was paid to careful and correct explanations in the proofs.

**Weaknesses:**

It is not clear how many interesting new cases these theorems cover. I did not feel the results and proofs were particularly surprising. For example, the ReLU proof quite heavily (but skillfully) relies on previously used enumeration of hyperplane partitions. And the tree structure in the ReLU theorem does not look very relevant to practical architectures.

**Questions:**

Line numbers are for the extended version from the supplement.

* I admit this is a matter of opinion, but the title seems a bit over the top. I would suggest to make it more informative.

* remark after lemma 3, I think it is great that you worked out the details of this enumeration problem. But as a point of feedback, parts of this remark read to me as unnecessarily hostile.

* line 372, typo "we the set"

* Corollary 6, technically you never defined "MIXED-NNT".

* proof of Theorem 12, you call inputs/outputs $x$ and $y$, but then you switch to $n$ and $o$. also in paragraph starting in line 547, I would double-check the transposition symbols.

* proof of Theorem 13, I found it frustrating that $D_i$ and $D_j$ denote very different objects in this proof.

**Limitations:**

see above

---

> ### Author Rebuttal · Authors · 2023-08-08
>
> Regarding the perceived weaknesses of our results, the class of networks where our algorithmic upper bound applies can indeed be considered as restrictive, but from a fundamental network-structural viewpoint we believe them to be a significant step beyond the previous state of the art. All results to date simply seem to suggest that solving NNT is a very difficult problem from the complexity-theoretic viewpoint, and in view of the strong existing lower bounds it is perhaps unrealistic to expect tractability for very general classes of networks (see also the comments of Reviewer 3t5Y).
>
> Thank you for all the individual comments; we provide responses to these below.
> 1) If the reviewers believe the title to be too strong or unspecific, we will of course change it in the final version.
> 2) We never intended this to sound hostile, but after reading the text again we understand that it could have come across as such. We will carefully reformulate this part to fix this.
>
> Points 3-6) will be fixed in the final version.

---

> > ### Comment · Reviewer_t5Nf · 2023-08-10
> >
> > Thank you for your reply and for addressing my and other reviewers' comments. I want to see if there is more discussion unfolding, but I am tentatively leaning towards increasing my score.

---

> > > ### Comment · Reviewer_t5Nf · 2023-08-17
> > >
> > > Dear authors,
> > >
> > > I am sorry for one more question late in the review stage. The question is about your "blow-up theorem" (Theorem 10).
> > >
> > > Consider any instance $D$ with distinct inputs in $\mathbb{R}^n$ and outputs in $\mathbb{R}$. Am I correct you can construct a network with 1) two hidden layers 2) satisfying out-degree 1 property and 3) perfectly fitting $D$, as follows: For each input $x_i\in D$ there are $O(n)$ ReLU neurons in the first hidden layer and one ReLU neuron in the second hidden layer, such that the function computed by the second layer neuron is indicator of $x=x_i$.
> > >
> > > Do you have thoughts about this construction and Theorem 10? (I'm sorry for any mistakes on my side.)
> > >
> > > [EDIT: I suppose the answer is that the size of your blowup does not depend on $|D|$. Sorry, I should have thought more before sending the question!]

---

> > > > ### Author Response · Authors · 2023-08-18
> > > >
> > > > Indeed, the difference is that in Theorem 10, the size of the obtained
> > > > blow-up depends *only* on the size of the original architecture -
> > > > which means that if the original architecture had size bounded by a
> > > > constant, the new one does as well (regardless of the number of
> > > > elements in $D$). This, in turn, allows us to solve the NNT problem on
> > > > the obtained instance while being guaranteed that this new
> > > > architecture will admit a solution that's at least as good as a
> > > > hypothetical optimal solution on the original architecture. (See also
> > > > the last sentence in Section 5.)

---

### Official Review · Reviewer_SU6D · 2023-07-07

**Soundness:** 4 excellent
**Presentation:** 3 good
**Contribution:** 2 fair
**Rating:** 6
**Confidence:** 3

**Summary:**

This paper studies the computational complexity of training linear and ReLU neural networks. Under assumptions such as the squared loss and the out-degree of every hidden neuron being exactly one, the authors prove that there exists an algorithm such that the global optimal solution can be computed in polynomial time for the linear and ReLU networks when the input dimension and the number of hidden neurons are fixed. To better understand the tractability of training linear networks, the idea of untangling is proposed and used to develop the existence of a polynomial-time algorithm for training linear networks. Although deciding whether a given architecture has an untangling is NP-hard in general, the authors prove that the complexity of determining this property for linear networks is linear time. They also provide a complexity result for computing an untangling of an architecture of bounded treewidth.

**Strengths:**

The novelty of this paper is clear. First, it ensures the existence of a training algorithm for linear and ReLU networks that achieves global optimal solutions in polynomial time under a different assumption compared to previous work. Second, the idea of untangling is proposed as a new assumption for studying these networks. These contributions could open up a new angle for studying the training complexity of ReLU and linear networks. The paper is well-written, and I enjoy reading the paper.

**Weaknesses:**

In my view, the assumptions are the main weaknesses of this work. The out-degree being 1 and the untangling of architecture are unrealistic assumptions. Deep neural network models in practice do not satisfy these assumptions. Second, finding the global optimal solution seems to be unnecessary. Usually, we would avoid getting to the global optimal solution because of overfitting. Third, when a machine problem is given, we only have the dataset so the number of hidden neurons $w$ is unknown. Based on these reasons, it seems hard to apply the results in this paper to applications.

**Questions:**

1. The squared loss is assumed in this work. Would it be possible to extend it to convex loss functions?
2. Usually, there are symmetries in the dataset. Would it be possible to leverage the symmetries or redundancies in the data and reduce the bounds? Would it be possible to apply that to drop some strong assumptions on the architecture and still maintain the polynomial complexity?
3. Do you need the number of bits of the input in the bounds? What computational models are you assuming?


**Limitations:**

The authors point out two questions at the end as possible future directions. However, due to the unrealistic assumptions mentioned above, there are strong limitations to applying the results to applications. Although this is a purely theoretical paper, I still think it would be a good idea to describe possible workarounds of the assumptions and how the current results could be impactful to machine learning problems.

---

> ### Author Rebuttal · Authors · 2023-08-08
>
> We are indeed well aware that the tractability results obtained in the article are of theoretical interest, and that real-life networks are unlikely to satisfy the stated structural properties. Even the tractability for general bounded-size ReLU-NNT architectures obtained in Section 5 are not likely to be of practical significance. Instead, the article follows up on a fundamental and well-established line of work that aims at understanding the theoretical boundaries of tractability (in the complexity-theoretic sense) for Neural Network Training.
>
> Regarding the three questions:
> 1) The main bottleneck for such an extension lies in solving the resulting convex program; in the case considered in the article (the commonly used squared loss function), this is a constrained least-squares problem, which is polynomially solvable. In the more general convex case, this still seems possible, but the arguments become more intricate and would involve considerations about machine precision and approximation that would obscure the main conceptual message. We will add a corresponding remark to the paper.
> 2) Indeed, it might be possible to study how the structure of the training set itself affects the complexity of the problem. However, to get rid of the exponential dependencies in the architecture parameters, one would have to show that the known hardness results (for an arbitrary training set) weaken or collapse when assuming specific structure. It is an interesting question that could point towards follow-up work.
> 3) This is indeed an oversight on our part. We’re assuming a standard Word-RAM model of computation, and the bit complexity does enter in our bounds in the same way that it does for the quadratic optimization methods used; we are grateful to this reviewer and Reviewer 3t5Y for pointing this out. We will of course update the running time bounds in the final version, and remark that this minor change (by a polynomial factor) does not affect the article’s contribution.
>
> Finally, we will expand the conclusion by highlighting that it would be useful to understand how one could avoid the strict prerequisites for our tractability results in more practical settings.

---

> > ### Comment · Reviewer_SU6D · 2023-08-14
> >
> > I would like to thank the authors for their response. Unfortunately, my concerns are not fully addressed after reading all the reviews and rebuttals. Having an out-degree of 1 is unrealistic so it seems hard to make the results useful. The complexity of blowing up a network is too high and I tend to agree with Reviewer Yud1 that treating the depth and width as constants seems to avoid the main difficulty of the problem. I also agree with the assessment by Reviewer t5Nf that the title needs to be changed to deliver the essence of the contribution. Given these factors, I will keep my rating unchanged. I'm happy to increase my rating if the authors can address my concerns.

---

### Official Review · Reviewer_Yud1 · 2023-07-19

**Soundness:** 2 fair
**Presentation:** 3 good
**Contribution:** 2 fair
**Rating:** 5
**Confidence:** 4

**Summary:**

This paper investigates the theoretical complexity of finding the global optimal solution of ReLU networks and linear networks. Specifically, it first considers a specially designed neural network, each hidden neuron of which has only one outgoing edge. By using the homogeneity of the ReLU function, it reparameterizes this network such that all the weights (except the first layer ones) are either -1 or +1 (i.e., moves the absolute values into the first layer weights). In this way, it shows that it is possible to find the global optimal by enumerating the partitions of the data space, with time complexity that is polynomial in the dataset size (but not polynomial in network size). It also shows that a normal fully-connected neural network can be “blown up” to one with the above special design (out-degree = 1 for each neuron), by replicating hidden neurons that have multiple out-going edges.

**Strengths:**

The obtained time complexity of finding the global optimal is polynomial in the dataset size.

This paper handles multi-layer ReLU/linear neural networks, while prior works only obtain similar results on one-hidden layer networks.

**Weaknesses:**

The main trick of this paper is to consider those variables (e.g., network width, depth, input dimension etc) in which the time complexity is exponential or greater as constants, so that the time complexity can be *stated as* polynomial – in the remaining variable (i.e., dataset size). I don’t think this is a clever trick, as it does not solve the problem at all. Note that the complexity is far greater than polynomial (or even than exponential) in the other variables, for width $w$ it is $|D|^{w2^w}$, and for depth it is larger than $|D|^{2^{\lambda^l}}$.

Given this large complexity in width and depth, the result of time complexity has little to no usefulness for practice.

The paper only focuses on the time complexity, while ignoring the space complexity – memory costs. Note that the blowing up procedure requires much more space complexity, which is exponential in depth. Recall that a neural network without blowing-up only requires space which is linear in depth. This huge memory cost is a drawback of the analyzed training algorithm, and should be considered and analyzed.

**Questions:**

No further questions

**Limitations:**

limitation is not discussed in the main text

---

> ### Author Rebuttal · Authors · 2023-08-08
>
> We would like to point out two potential misunderstandings related to the purpose and scope of the article. Indeed, we do obtain polynomial-time algorithmic upper bounds under the assumption that certain parameters of the input (such as the size of the architecture) are fixed to be constants. However, as previous work has shown and as has been elaborated on in depth in the paper, this is mathematically proven to be simply unavoidable (see the discussion on lines 191-204); in this sense, our results even provide essentially optimal running time upper-bounds. The emphasis of the paper is very clearly positioned on pushing the complexity-theoretic frontiers of tractability, and it most certainly is not our intention to generate the impression of attempting to further the state-of-the-art of Neural Network Training in practice.
>
> The second point of concern - and perhaps the more severe misunderstanding - lies in the characterization of our algorithmic results as a “trick” where one simply hides the exponential dependencies in the algorithms. This is not the case; the article is very explicit concerning which parts of the input are considered to be fixed. Crucially, there exists a vast literature on complexity-theoretic tractability results on problems that focus entirely on the theoretical study of the boundaries of tractability under the assumption of certain “variables” of the input being fixed. Not only are there many such publications at NeurIPS and related venues every year (see the discussion of related work in the second paragraph of the article), but this fundamental idea is pervasive throughout computer science and in fact forms the central concept of the subfield of parameterized complexity. In fact, the criticism raised here could be used to automatically dismiss, among others, each of the articles cited in that paragraph, and hence we believe it to be unjustified.
>
> Regarding the final point, while a detailed space-complexity analysis is beyond the scope of the paper, we do agree that it is interesting and will make sure to mention the fact that the blow-up procedure has an impact on the space complexity of the problem in the final version.

---

> > ### Comment · Reviewer_Yud1 · 2023-08-13
> >
> > Thanks for the reply. However, my major concern about the validity of fixing certain variables, especially these fixed variables turns out to be exponential, still remains. I will discuss with AC and other reviewers about this concern.

---

> > > ### Author Response · Authors · 2023-08-15
> > >
> > > Thank you for letting us know of your concerns. In order to substantiate our claim that it is a standard practice to analyze problems under the assumption that certain variables of the input are fixed, we provide below an explicit list of recent references – all taken from the article – which employ this same assumption. Naturally, there are many, many more examples of this beyond the references we explicitly cite in our article.
> > >
> > > Abrahamsen, M., Kleist, L., and Miltzow, T. Training neural networks is er-complete. NeurIPS 2021
> > >
> > > Arora, R., Basu, A., Mianjy, P., and Mukherjee, A. Understanding deep neural networks with rectified linear units. ICLR 2021
> > >
> > > Boob, D., Dey, S. S., and Lan, G. Complexity of training relu neural network. Discret. Optim. 44
> > >
> > > Brand, C., Koutecky, M., and Ordyniak, S. Parameterized algorithms for milps with small treedepth. AAAI 2021
> > >
> > > Dahiya, Y., Fomin, F. V., Panolan, F., and Simonov, K. Fixed-parameter and approximation algorithms for PCA with outliers. ICML 2021
> > >
> > > Dey, S. S., Wang, G., and Xie, Y. Approximation algorithms for training one-node relu neural networks. IEEE Trans. Signal Process. 68
> > >
> > > Froese, V., Hertrich, C., and Niedermeier, R. The computational complexity of relu network training parameterized by data dimensionality. J. Artif. Intell. Res. 74
> > >
> > > Ganian, R. and Korchemna, V. The complexity of bayesian network learning: Revisiting the superstructure. NeurIPS 2021
> > >
> > > Ganian, R., Kanj, I. A., Ordyniak, S., and Szeider, S. Parameterized algorithms for the matrix completion problem. ICML 2018
> > >
> > > Ganian, R., Hamm, T., Korchemna, V., Okrasa, K., and Simonov, K. The complexity of k-means clustering when little is known. ICML 2022
> > >
> > > Goel, S., Klivans, A. R., Manurangsi, P., and Reichman, D. Tight hardness results for training depth-2 relu networks. ITCS 2021
> > >
> > > Gruttemeier, N. and Komusiewicz, C. Learning bayesian networks under sparsity constraints: A parameterized complexity analysis. J. Artif. Intell. Res. 74
> > >
> > > Simonov, K., Fomin, F. V., Golovach, P. A., and Panolan, F. Refined complexity of PCA with Outliers. ICML 2019

---

> > > > ### Comment · Reviewer_SU6D · 2023-08-15
> > > >
> > > > I would like to point out that some of the papers provided above have other major contributions. For example, the paper by Arora et al. (2018) provides other theorems characterizing the piecewise linear representation of ReLU networks. They only have a section dedicated to RELU-NNT. Since this paper entirely focuses on new algorithmic upper bounds, it is important to properly describe the bounds. For instance, the claim "...an arbitrary bounded-size ReLU-activated deep neural network architecture ...guaranteed to admit polynomial-time training" in Line 305 should be improved to ensure that the bound is only polynomial in the number of data points when treating all the other parameters constants.

---

> > > > > ### Author Response · Authors · 2023-08-16
> > > > >
> > > > > To address this concern, for the final version we will carefully go over the claims made regarding the upper bounds and check to make sure they cannot be accidentally misinterpreted in a way which would ignore the preconditions for those statements (e.g., assuming the architecture to be of constant size). We expect that this will simply entail expanding and elaborating on the adjective "bounded-size", which we consistently used in such claims with the intention of conveying this information.
> > > > >
> > > > > We remark that even if a reader were to accidentally misinterpret such claims in the explanatory text, the intended meaning could always be recovered from the corresponding formal statements (the example mentioned in the comment is formalized in the theorem statement on lines 331-335). But we will leave no room for misinterpretation in the final version.

---

### Author Rebuttal · Authors · 2023-08-08

We thank all reviewers for their feedback and insightful comments. Detailed responses to individual questions and potential misunderstandings are provided as separate comments to each review.

---

### Decision · Program_Chairs · 2023-09-21

**Decision:**

Accept (poster)

**Comment:**

The paper provides new computational complexity upper bounds for empirical risk minimization (ERM) of special families of neural networks. This problem is well-known to be NP-hard in the general case. The main contribution is to isolate special cases where polynomial time algorithms for ERM are possible:

* constant-size ReLU neural networks with hidden neurons having out-degree one;
* linear neural networks that obey a property called "untangling"; if the networks have constant treewidth, this property can be verified in linear time.

Most reviewers appreciated the technical contributions of the paper. However, during the discussion phase, concerns were raised about the utility of the bounds (technically, the algorithms are only fixed-parameter tractable). The issue of memory cost blow-up was also raised. The authors are encouraged to highlight (and address) these limitations while preparing the final version.